# Curve Similarity Model for Real-Time Gait Phase Detection Based on Ground Contact Forces

**DOI:** 10.3390/s19143235

**Published:** 2019-07-23

**Authors:** Huacheng Hu, Jianbin Zheng, Enqi Zhan, Lie Yu

**Affiliations:** 1School of Information Engineering, Wuhan University of Technology, Wuhan 430070, China; 2School of Electronic and Electrical Engineering, Wuhan Textile University, Wuhan 430073, China

**Keywords:** ground contact forces, force sensitive resistors, curve similarity model, threshold method, similarity distance

## Abstract

This paper proposed a new novel method to adaptively detect gait patterns in real time through the ground contact forces (GCFs) measured by load cell. The curve similarity model (CSM) is used to identify the division of off-ground and on-ground statuses, and differentiate gait patterns based on the detection rules. Traditionally, published threshold-based methods detect gait patterns by means of setting a fixed threshold to divide the GCFs into on-ground and off-ground statuses. However, the threshold-based methods in the literature are neither an adaptive nor a real-time approach. In this paper, the curve is composed of a series of continuous or discrete ordered GCF data points, and the CSM is built offline to obtain a training template. Then, the testing curve is compared with the training template to figure out the degree of similarity. If the computed degree of similarity is less than a given threshold, they are considered to be similar, which would lead to the division of off-ground and on-ground statuses. Finally, gait patterns could be differentiated according to the status division based on the detection rules. In order to test the detection error rate of the proposed method, a method in the literature is introduced as the reference method to obtain comparative results. The experimental results indicated that the proposed method could be used for real-time gait pattern detection, detect the gait patterns adaptively, and obtain a low error rate compared with the reference method.

## 1. Introduction

Gait pattern detection is an effective way to monitor and analyze the condition of human walking [1]. Gait analysis plays an important role in the fields of lower limb rehabilitation robot, assisting exoskeleton human–machine coupling systems, and daily life assistance [2]. On the basis of the specific objective of gait analysis, various sensor platforms and algorithms have been developed to classify all or some gait phases, including force sensors [3,4,5,6], inertial sensors [7,8,9], foot switches [3], and electromyography (EMG) sensors [10]. 

Recently, a large number of new sensors and their signal processing methods have been used for gait recognition. A gait recognition system in loose clothing has been proposed in the works of [11,12], where four sets of flexible piezoelectric sensors were embedded in both knee and hip parts on the patient’s pants. Mileti et al. designed a pair of shoes with an Inertial Measurement Unit (IMU) and foot switches on each foot for gait quality assessment [13]. Zhou et al. combined color sensors, depth sensors, and inertial sensors for gait data collection and gait recognition [14]. Among the sensor platforms, force sensor platforms, such as load cell, which could be located in shoe soles to measure the ground contact force (GCF), may represent the gold standard method for gait analysis [15]. The electrical resistance change of a load cell is proportional to the GCF acted by the human foot. As reported in another paper [16], the measuring value changes of load cell could be correlated directly to the gait patterns because each gait pattern has a unique GCF pattern.

In order to achieve gait analysis, different sensor fusion and machine learning methods have been presented, such as fuzzy logic [17], support vector machine (SVM) [18], hidden Markov models (HMM) [19,20], naive Bayes classification (NBC) [16], neural network (NN) [17,21], and Gauss mixture model (GMM) [18]. Tian [22] uses the VICON MX 3D gait system to collect the joint angle values of the hip, knee, and ankle in these five gaits and performs gait recognition based on coalitional game-based feature selection and extreme learning machine (ELM).

Generally, the division of gait phases is based on the threshold method, which sets a threshold to divide the GCF into on-ground and off-ground statuses. Catalfamo et al. [23] and Lopez-Meyer et al. [6] calculate the threshold with the maximum and minimum GCFs of gait cycles, which meant that the GCF should be post-processed after data acquisition. Mariani et al. [5] used 5% body weight for threshold computation with the result that the weight of each subject should be obtained prior to the experiments. Lie Yu et al. [24] and Jing Tang et al. [25] declared that the methods in the works of [5,23] were not adaptable to different people and different walking speeds. The faster the same person walks, and the larger the GCF value, it is difficult to achieve real-time gait phase accurate detection by simply using the threshold method. Therefore, Yu et al. proposed the proportional method (PM), which calculates the sums and proportions of GCFs for gait pattern detection [24]. Two proportional factors were optimized and selected for all subjects in all experiments, and this PM achieved a high average reliability. Tang et al. presented the self-tuning triple-threshold algorithm (STTTA), which searched out GCF extremes in real time for threshold computation [25]. Three initial threshold values were set for all subjects in all experiments, while three adjustable thresholds would be obtained to adapt the human walking with a detection accuracy of almost 90%. According to Yu and Tang’s research results, the accuracy of real-time gait phase detection is close to 90%, but in the actual exoskeleton robot cooperative control, especially in the process of weight bearing and speed varying walking, the accuracy of gait phase detection could not meet the real-time control requirements of the robot. Meanwhile, some people would land first on their heels, while others would land first on their balls, resulting in simple proportional methods finding it difficult to adapt to the different types of walking patterns of people.

The GCF signals of time series are considered as a curve, which means that the GCF points are considered as a function of time. In view of this, this paper proposes a gait pattern detection method based on the curve similarity model (CSM), and the GCF points in a time interval are taken as a curve to study. In the gait cycle, the GCF curve over a period of time can distinguish between different gait patterns. In particular, the pattern of GCF curve changes during gait phase switching can be used as an indicator to detect the next new gait phase. This paper presents a gait detection mode method based on the curve similarity model, including the definition of curve similarity, gait curve template, and its construction and application in gait detection. Zheng et al. [26] defined the curve similarity using the absolute value integral of the difference between two functions as the distance measure of two curves. In this paper, the curve similarity model is used to realize the detection of specific gait patterns. Specifically, the curve is composed of a series of discrete ordered GCF data points, and the CSM is built by a small amount of offline data in order to train two kinds of curve templates, which are used to detect off-ground and on-ground statuses, respectively.

Similarly, in real-time gait detection, the GCF signal at the current and past intervals is used as a test curve, and is continuously compared with the gait template curve to calculate the similarity distance between the test curve and the template curve. If the similarity distance is smaller than a given threshold, the state represented by the gait template curve is the gait state at the current time and the subsequent time until the next gait state is detected to change.

Finally, the gait patterns could be differentiated according to the status division based on the detection rules. In order to test the detection error rate of the proposed method, a method in the literature is introduced as the reference method to obtain comparative results.

## 2. Method

### 2.1. Subjects and Procedures

This study included twenty-four subjects of average age 23.1 ± 3.2 years and average mass 68.2 ± 7.6 kg with no history of foot diseases or limitations. The study was approved by Beijing Sport University Institution Review Board (Re.-No. 2019007H). Before the experiments, the subjects gave their written informed consent for participation in this study as the purpose was explained in detail to each of them and their safeties could be ensured.

To validate our research, a gait phase detection system was designed as pictured in Figure 1. Two sets of load cells (LOSON LSH-10, LOSON Instrumetation, Nankin, China) were severally located in the insole of ball and heel. The load cell signals were collected at a sample frequency of 2000 Hz through 16-bit AD converters. A pressure tester (TLS-S1000W, Jinan Zhongchuang Industry Test System Co., Ltd., Jinan, China) is used to calibrate the load cell such that the amplified output signal within a range of 0–5 V correlates with the measured mass of 0–200 kg [27]. After data acquisition and sensor calibration, the GCF signals were filtered by average filter with a cut-off frequency of 100 Hz to eliminate the unnecessary high frequency noise.

Each subject was asked to perform five trials to wear the designed shoes (i.e., seen in Figure 1) to walk on a treadmill. The five trials were performed for 30 s each at a designated constant speed of 2 km/h, 3 km/h, 4 km/h, 5 km/h, and 6 km/h in turn. According to the adult’s step size of 60 cm~80 cm, the duration of each gait cycle at a speed of 2~6 km/h is about 1000~2000 ms. The slower the speed, the longer the gait cycle. Each subject took an average of 110 steps at different speeds with 150 s. Using the same acquisition device, we selected 10 subjects from the above-mentioned 24 subjects for continuous shift 60 s walking experiments, using real-time data for online gait phase detection.

### 2.2. Gait Pattern Detection Algorithm

Generally, a complete gait cycle could be divided into two main phases, that is, stance phase and swing phase. As two sets of load cell were mounted in each shoe, four types of gait patterns could be differentiated. Specifically, when the foot fully contacts the ground, the gait pattern is the stance phase. When the foot totally leaves the ground, the gait pattern is the swing phase. The transitions between the stance and swing phases are gait events. When the ball leaves the ground and the heel contacts the ground, the gait pattern is heel strike. When the ball contacts the ground and the heel leaves the ground, the gait pattern is heel-off.

Usually, the division of contacting or leaving the ground is made through setting a threshold. Several drawbacks existed for threshold-based methods, which have been summarized in detail in the works of [21,28]. To solve these problems, this paper used the curve similarity model to implement the status division of contacting or leaving the ground.

If this curve similarity model is built, a curve consisting of several GCF points would be discriminated to be on-ground or off-ground status. The key to this model is to determine the start point of each status so that the GCF will be discriminated to be in the same status for a subsequent period of time until the next different gait starting point is detected. Meanwhile, if more GCF points are used, the curve similarity model would gain better results. However, more computation resource would be costed. In this paper, four GCF points are utilized to build the curve similarity model. 

#### 2.2.1. Global Threshold Method and Its Disadvantage

By setting a threshold *Thr*, GCFs can be divided into on-ground and off-ground statuses:(1)$$G(i)=\left\{ \begin{array}{l} 1,T(i)\geq Thr\\ 0,T(i)<Thr \end{array}\right.$$
where *T*(*i*) is the GCF at *i*-th point and *T**hr* is the threshold.

The Lopez–Meyer [6] method calculates the mean values of the maximum and minimum threshold of GCFs as the *Thr* for gait detection. The mathematical expressions are as follows:(2)$$\left\{ \begin{array}{l} T_{MAX}=\frac{1}{n}\sum_{i=1}^{n}T_{\max}(i)\\ T_{MIN}=\frac{1}{m}\sum_{i=1}^{m}T_{\min}(i)\\ Thr=T_{Lopez}=\alpha \times (T_{MAX}-T_{MIN})+T_{MIN} \end{array}\right..$$

The scale factor is obtained by the leave-one-out method, which is used in gait detection system. In this paper, set α = 0.094. Usually, *m* is not equal to be *n*. At the most, the GCF has only one maximum and only one minimum value in a complete gait cycle. However, an incomplete gait cycle would lead to only one *T*max or *T*min. Therefore, an abnormality will occur in the gait discrimination.

Although the threshold value is intuitive, the sensor detection pressure is related to the wearer’s weight and walking posture, and it is easy to generate abnormal values, which can be clearly seen in Figure 2. Therefore, when using the threshold method for gait detection directly, it is necessary to eliminate these abnormal conditions. If window filtering is used, the system detection delay will be increased in a real-time system.

The GCF value change is related to the walking speed. As seen in Figure 3, when the walking speed changes from 2 km/h to 6 km/h, the GCF value gradually increases, and the corresponding *Thr* also gradually increases. Therefore, for variable speed walking, the threshold of *Thr* is uncertain.

Combined with the above analysis, it can be seen that the threshold method is simple to calculate, but the stability and reliability need to be improved in real-time calculation. In the post-view after-the-fact analysis, combined with window filtering, it can be used as a reference for gait detection for comparative analysis.

#### 2.2.2. Starting Flag of on-Ground and off-Ground Statuses

The starting time of each status was determined by comparing GCFs with the threshold calculated through the Lopez–Meyer method [6]. 

Obviously, the threshold was only used in the training of the curve similarity model. As depicted in Figure 4, a curve consisting of four GCF points is obtained referring to the threshold. *x*(*i*) is GCF at the starting flag of on-ground status, while *x*(*i* − 1), *x*(*i* − 2), and *x*(*i* − 3) are the GCFs identified as off-ground status. If this curve could be recognized accurately, the on-ground status would be differentiated initially. The future GCFs with a long period after *x*(*i*) would also be judged to be on-ground status. Meanwhile, if the ending flag of on-ground status could be recognized, the on-ground status would be differentiated totally.

However, the ending flag of on-ground status is extremely similar to the starting flag of off-ground status, as pictured in Figure 5. Obviously, a one point difference existed between the two flags. If the starting flag of off-ground status is identified, the ending flag of on-ground status could be deduced. As shown in Figure 5, a curve consisting of four GCF points is acquired referring to the threshold. *x*(*i*) is GCF at the starting flag of off-ground status, while *x*(*i* − 1), *x*(*i* − 2), and *x*(*i* − 3) are the GCFs identified as on-ground status. Similarly, if this curve could be identified accurately, the off-ground status would be differentiated initially. The future GCFs with a long period after *x*(*i*) would also be judged to be off-ground status. Additionally, the ending flag of off-ground status could be deduced according to the recognition of the starting flag of on-ground status, as demonstrated in Figure 4.

Thus, the status division could be made by recognizing the starting flags of on-ground and off-ground statuses according to Figure 4 and Figure 5. The curve similarity model is built to identify the two flags, which is described in the following section.

#### 2.2.3. Curve Similarity Model

Before building this model, a curve consisting of four GCF points should be constructed, noted as L=〈x(i),x(i−1),x(i−2),x(i−3)〉. However, according to actual experiments, if only four GCF points were used to establish the model, it would lead to many detection mistakes. Therefore, scale difference was added to extend this curve, which could be rewritten as follows:(3)$$L=\langle x(i),x(i-1),x(i-2),x(i-3),L^{\left(k\right)}\rangle ,$$
where $L^{\left(k\right)}$ is the *k*-scale difference of *L*; and the range of *k* is selected to be 1, 2, and 3 in this model. Then, $L^{\left(k\right)}$ could be expanded as follows:(4)$$L^{\left(k\right)}=\langle x^{\left(1\right)}(i),x^{\left(2\right)}(i),x^{\left(3\right)}(i),x^{\left(1\right)}(i-1),x^{\left(2\right)}(i-1),x^{\left(1\right)}(i-2)\rangle ,$$
where *x*^(*k*)^(*i*) = *x*(*i*) − *x*(*i* − *k*) is the *k*-scale difference of the sequence at *x*(*i*), and Equation (4) could be described as follows:(5)$$L^{\left(k\right)}=\langle \begin{array}{l} x(i)-x(i-1),x(i)-x(i-2),x(i)-x(i-3),\\ x(i-1)-x(i-2),x(i-1)-x(i-2),x(i-2)-x(i-3) \end{array}\rangle .$$

After curve extension, the model should be constructed and noted as *φ*(*L*), where *φ*(*L*) is the transformation function. Curve similarity theory is used to measure the degree of similarity between two curves. Firstly, given two curves, *L_A_* and *L_B_*, the similarity distance between them could be expressed as follows:(6)$$d=\left|\phi (L_{A})-\phi (L_{B})\right|,$$
where *d* is the similarity distance between two curves, and |●| is the absolute distance. For *d*, given a threshold *ε*, if *d* ≤ *ε*, the curves *L_A_* and *L_B_* are proven to be similar. Otherwise, they are not.

According to Equations (3) and (5), there are ten elements in curve *L*, including four GCF point and their scale differences. In order to describe the curve similarity model in an understandable manner, the curve *L* is rewritten as follows:(7)$$\left\{ \begin{array}{l} L=\langle x_{1},x_{2},\cdots ,x_{j},\cdots ,x_{10}\rangle \\ x_{1}=x(i),x_{2}=x(i-1),x_{3}=x(i-2),x_{4}=x(i-3),\\ x_{5}=x(i)-x(i-1),x_{6}=x(i)-x(i-2),x_{7}=x(i)-x(i-3),\\ x_{8}=x(i-1)-x(i-2),x_{9}=x(i-1)-x(i-3),x_{10}=x(i-2)-x(i-3) \end{array}\right..$$

To transform every element of *L* in the same dimension, Gaussian function is selected to be the transformation function *φ*(*L*), which could be described as follows:(8)$$\left\{ \begin{array}{l} \phi (L)=\langle \varphi (x_{1}),\varphi (x_{2}),\cdots ,\varphi (x_{j}),\cdots ,\varphi (x_{10})\rangle \\ \varphi (x_{j})=\exp(\frac{-{\left(x_{j}-\mu_{j}\right)}^{2}}{2\delta_{j}^{2}}) \end{array}\right.,$$
where *ϕ*(*x_j_*) is the transformation function of *x_j_*. Meanwhile, *μ_j_* and *δ_j_* are the average value and standard deviation of *x_j_*, respectively.

Thus, the gains of *μ_j_* and *δ_j_* should be calculated or trained from gait phase labeling data. For *L_A_* and *L_B_*, one of them is considered as a reference curve or template curve, which is trained and represents a classification, while the other is considered as a comparison curve to measure similarity of two curves. Assuming that *L_A_* is the template curve, the obtained *x_j_* for *L_A_* is equal to *μ_j_* by coincidence. Meanwhile, *L_B_* is the comparison curve. Then, the similarity distance between *L_A_* and *L_B_* could be rewritten as follows:(9)$$\begin{array}{l} d=\left|\phi (L_{A})-\phi (L_{B})\right|\\ =\left|\langle \varphi (\mu_{1},\mu_{2},\cdots ,\mu_{j},\cdots \mu_{10})\rangle -\langle \varphi (x_{1},x_{2},\cdots ,x_{j},\cdots x_{10})\rangle \right|\\ =\sum_{j=1}^{10}\left|\exp(\frac{-{\left(\mu_{j}-\mu_{j}\right)}^{2}}{2\delta_{j}^{2}})-\exp(\frac{-{\left(x_{j}-\mu_{j}\right)}^{2}}{2\delta_{j}^{2}})\right|\\ =\sum_{j=1}^{10}\left|1-\exp(\frac{-{\left(x_{j}-\mu_{j}\right)}^{2}}{2\delta_{j}^{2}})\right|\\ =10-\sum_{j=1}^{10}\exp(\frac{-{\left(x_{j}-\mu_{j}\right)}^{2}}{2\delta_{j}^{2}}) \end{array}$$

The purpose of building this curve similarity model is to measure the similarity of the reference curve and comparison curve. Once the comparison curve is similar to the reference curve, the comparison curve has the same classification as the reference curve. Given a threshold *ε*, if *d* ≤ *ε*, the curves *L_A_* and *L_B_* are proven to be similar. Generally, the threshold *ε* is selected as follows:(10)$$0<\frac{\varepsilon }{P}\leq 0.2,$$
where *P* is the length of *L*, and equals 10 in this model. Then, *ε* = 2 is preferable.

As depicted in Figure 4 and Figure 5, the starting flags of on-ground and off-ground statuses should be recognized. Therefore, two curve similarity models should be constructed severally for on-ground and off-ground statuses, noted as *φ*(*L*_1_) and *φ*(*L*_2_). Meanwhile, each model is built through Equation (8), and the parameters in these models are selected through Equation (10).

The testing data point is noted as the comparison curve *φ*(*L**c*) and we calculate the similarity distances as follows:(11)$$\left\{ \begin{array}{l} d_{1}=d\left(\varphi \left(L_{1}\right),\varphi \left(Lc\right)\right)\;\\ d_{2}=d\left(\varphi \left(L_{2}\right),\varphi \left(Lc\right)\right) \end{array}\right..$$

For *φ*(*L*_1_), if *d*_1_ is smaller than *ε*, the testing data point is classified as on-ground status. On the other side, for *φ*(*L*_2_), if *d*_2_ is smaller than *ε*, the testing data point is classified as off-ground status. However, if the testing data point fails to satisfy both conditions, the testing data point is classified as its last status.

### 2.3. Gait Phase Classification by GCF

As mentioned above, detecting the starting flag of off-ground status can determine whether the gait phrase enters off-ground status, and detecting the starting flag of on-ground status can determine whether the gait phase enters off-ground status. It is very difficult to capture solely the occurrence of the starting flags of two statuses because that the range of pressure varies greatly. The range before and after is used as the interval from which the starting flags of off-ground and on-ground statuses occur, and the new gait classification is redefined. 

In the process from on-ground to off-ground, when the GCF is less than the threshold pressure and before middle off-ground, we define the status as the initial status of off-ground, referred to as the initial off-ground. There is a corresponding process from off-ground to on-ground, which is called the initial on-ground. Its duration range is from the terminal off-ground to the first time when GCF is greater than the threshold pressure.

From a large number of walking gait pressure curves, this pressure change lasts from 40 to 60 ms, where the faster the walking, the shorter the duration. We take a continuously changing pressure curve as our research object, and develop the gait detection model within one gait cycle to four status detection, that is, the initial off-ground, the off-ground, the initial on-ground, and on-ground statuses order by order. Here, the on-ground status is labeled as 3, the initial off-ground status is labeled as 1, the off-ground status is labeled as 0, and the initial on-ground status is labeled as 2. In this way, we divide a normal walking process into four gait phases that are repeated over and over again, as shown in Figure 6.

Therefore, the specific classification of gait phase is as follows:(1)Traverse a piece of GCF data, calculate *Thr*;(2)In each gait cycle, when the gait motion moves from on-ground to off-ground status, the first data point whose GCF is less than *Thr* is calculated as the starting position of the initial off-ground status, and then the delay time *t_W_* where the data point is the ending position of the initial off-ground status, and the subsequent point is taken as the off-ground status;(3)Likely, when the gait motion moves from off-ground to on-ground status, the first data point whose GCF is greater than *Thr* is calculated as the ending position of the initial on-ground status, and then time *t_W_* in advance where the data point is the starting position of the initial on-ground status, and the subsequent point is taken as the on-ground status;(4)On the part of abnormal gait tagging data, such as gait mutation, *t_W_* is too large or too small, take the form of manual updating.

Usually, set *t_W_* = 40~60 ms, here *t_W_* = 50 ms, if one gait cycle is 1000 ms~2000 ms, the maximum theoretical error of detection is 2.5%~5%.

### 2.4. Evaluation Protocols of the Gait Phase Detection

In order to evaluate the reliability of the proposed method, the reference method should be determined. As reported in the work of [6], the Lopez–Meyer method was approved through comparing itself with the “GAITRite system”, and gained a comparative and reliable confidence of 95%. Therefore, the Lopez–Meyer method was introduced as the reference method. However, the term of detection error rate was used to replace the reliability. Then, the detection error rate of this study was determined by comparing the detection results between the reference method and the proposed algorithm.

Four points are taken as a gait curve, and gait data is traversed. Each training sample contains the GCF values of four data points as input, and the gait phase classification of the last data point is used as the classifier output.

In the evaluation protocol of actual detection statuses, the corresponding detection results under each status are as described in Table 1.

The symbol “*” indicates no evaluation. In a gait movement, the actual number of gait phases is *cn_i_*. The number of corresponding gait phases detected each time is *dn_i_*, where the correct detection number is *tn_i_*, the missed detection number is *mn_i_*, the false detection number is *fn_i_*, and the over detection number is *sn_i_*, where *i* = 1 or 2.

Then, the total error values for gait detection of off-ground and on-ground statuses are defined as follows:(12)$$\begin{array}{l} F_{1}=\left(mn_{1}+fn_{1}\right)/2\\ F_{2}=\left(mn_{2}+fn_{2}\right)/2 \end{array}$$

Obviously, *tn**_i_* = *cn**_i_*− *mn**_i_*. At the same time, *mn_i_* is called the number of missed detections, *fn_i_* is also called the number of false detections, and *sn_i_* is called the number of over detections. When the number of training samples is fixed, the smaller the *F*_1_ or *F*_2_, the lower the error rate, the higher the accuracy of the system, and the better the corresponding curve template.

In the experiments, we mainly evaluate gait detection to solve a classification problem. As a classification problem, we use classification error rate as a measurement of performance evaluation. The error rate of each gait phase detection for off-ground and on-ground status, and the total error rate are defined as

(13)$$\begin{array}{l} E_{1}=\left(mn_{1}+fn_{1}\right)/cn_{1}\times 100\%\\ E_{2}=\left(mn_{2}+fn_{2}\right)/cn_{2}\times 100\%\\ E_{3}=\left(mn_{1}+fn_{1}+mn_{2}+fn_{2}\right)/(cn_{1}+cn_{2})\times 100\% \end{array}$$

At the same time, we also use the method from the literature [27,28] to evaluate the accuracy of gait phase detection.
(14)$$\begin{array}{l} E_{4}=\sum_{k=1}^{N}h_{k}/N\times 100\%\\ h_{k}=\left\{ \begin{array}{l} \begin{array}{cc} =1 & c(k)=0,h(k)=0 \end{array}\\ \begin{array}{cc} =1 & c(k)=1,h(k)=1 \end{array}\\ \begin{array}{cc} =0 & other \end{array} \end{array}\right. \end{array}$$
where *c*(*k*) is the true gait phase represented by every GCF data point, *h*(*k*) is the result of gait phase detection for every moment, and *N* is length of all data points. *E*_4_ can be considered as the total data point error rate.

### 2.5. Development of Gait Phase Classifiers by EC

Note that the quality of curve template is a very important index for gait phase detection. A good curve template will be able to better distinguish the starting flag of off-ground and on-ground status. Therefore, the key is to find the optimal parameters of the template curve. We label the collected gait data for four categories by the Lopez–Meyer method and divide them into a training set and a testing set. In this paper, an evolutionary computation (EC) algorithm is presented to carry out random intelligent search. Because the off-ground and on-ground curve templates are independent of each other, the evolutionary computation will be carried out twice.

The overall idea of evolutionary computation is to generate random *PopSize* individuals *S* = {*S*_1_, *S*_2_, … *S_PopSize_*}, each of which corresponds to a set of template curve parameters *μ* and *δ*.

In each iteration search of evolutionary computation, each individual will correspond to a total detection error value *F*_1_ or *F*_2_ as Formula (9), where the smaller the total detection error value, the better the individual. Fitness value is calculated by the sum of all the error values of the training dataset at one iteration.

The specific Algorithm 1 steps are described as follows:
**Algorithm 1.** Optimal calculation of curve parameters for CSM**Input**: training sample *X*, the size of individuals *PopSize*, and the maximum number of generations *MaxGen***Output**: template curve parameters *μ* and *δ*1: initialization, randomly generate initial populations *S*2: **for**
*j* = 1 to 2**PopSize* do3:  *S*[*j*]. *μ and*
*S*[*j*]. *δ*←rand()4:  evaluate the fitness value *F*_1_ or *F*_2_ based on Formula (12) with *X*5: **end for**6: **for**
*k* = 1 to *MaxGen*7:   sort *S*[1:*2*PopSize*] according to fitness8:   classify *S*[1:*PopSize*] into four categories according to fitness9:   *S[*1:*PopSize/4**].kind=1, S[PopSize/4**:PopSize/2**].kind=2*10:   *S[PopSize/2**: PopSize*3/4**].kind=3, S[PopSize*3/4**:PopSize**].kind=4*11:   **for**
*i* = 1 to *PopSize*12:   produce *kind* + 1 descendants *SubS* from *S*[*j*] and *kind*=*S*[*j*].kind13:   evaluate the fitness value *F*_1_ or *F*_2_ of each *SubS* based on Formula (12) with *X*14:   sort *SubS* according to fitness15:   *S*[*j + PopSize*]←*SubS*[0] 16:   **end for**17: **end for**18: **return**
*μ* and *δ* from *S*[0]

According to the maximum number of generations *MaxGen* as termination conditions.

Normally, *PopSize* = 20, *MaxGen* = 200.

### 2.6. Real-Time Gait Detection by CSMs

In real-time gait detection, we directly use gait detection by trained two CSMs from multiple people and different speeds, as shown in Figure 7. Taking the left foot as an example, the real-time gait detection Algorithm 2 is as follows:
**Algorithm 2.** The real-time gait detection by CSMInput: the pressure curve *X*(*k*), the template curves *L*_1_ and *L*_2_, the threshold ε = 2Output: real-time gait detection results *G*(*k*) 1: system initialization, *G*(1) = 1, *G*(2) = 1, *G*(3) = 1, *k* = 4, the gait is in the ground state2: **repeat**3:  *L_c_*←*X(k)* = [*x_k−_*_3_, *x_k−_*_2_, *x_k−_*_1,_
*x_k_*] 4:  calculate *d*_1_ = *d*(*φ*(*L*_1_), *φ*(*L**c*)) and *d*_2_ = *d*(*φ*(*L*_2_), *φ*(*L**c*)) 5:  **If**
*d*_1_ ≤ ε6:   G(*k*) = 17:  **elseif**
*d*_2_ ≤ ε8:   G(*k*) = 09:  **else**10:   G(*k*) = G(*k* − 1) 11:  *k* = *k* + 112: **until** system stop

## 3. Results and Discussion

There were two different databases for the experiment; one database comes from 24 subjects including 120 sample files and the other database came from 10 subjects including 10 sample files. In each sample file, the global threshold method described in Section 2.3 was used to label the data, and the data of each gait cycle were labeled as four gait phases. Walking at speeds of 2–6 Km/h for 30 s, a total of 100 gait cycles around would be obtained for each subject.

Because the adoption rate was 100 Hz, labeled gait classification data of 3000 rows would be generated for training and testing by traversing each sampling point in each sample of first database. Therefore, our gait classification records for the whole training and testing were about 120 × 3000 = 360,000.

For the first database, we selected some samples as training set and the rest as a testing set. Firstly, 25, 50, and 75 samples from subjects #1~5, #1~10, and #1~15, respectively, were selected as training set to verify the accuracy and reliability of the model. Secondly, five samples from subjects #1, #4, #9, #12, and #15 were selected as a training set, and the all subjects were selected as a testing set to verify the robustness of the model. For each training set, we repeated it 10 times to obtain different models to verify the reliability and stability of the models. Finally, for the second database, we chose the model trained to test the generalization performance of the model for continuous variable speed walking.

### 3.1. Results of Gait Pattern Detection

We used 50 samples from 10 subjects as training data and the rest of the samples as testing data. The ten elements of *L* in each gait cycle could be obtained by referring to the threshold calculated through the Lopez–Meyer method. Then, mean value and standard deviation for each element could be calculated. *μ* and *δ* from the curve similarity model for off-ground and on-ground status are listed in Table 2 and Table 3, respectively.

In the first experiment, with 50 samples of the training set, the EC algorithm searches the optimal parameters of two CSMs for off-ground and on-ground status, respectively, to minimize the corresponding *F*_1_ and *F*_2_ values. *μ* and *δ* from two CSMs for off-ground and on-ground status are listed in Table 2 and Table 3, respectively.

Then, the training sets were processed using the selected coefficients to obtain the detection error rate. Meanwhile, the testing sets from the remaining subjects were processed using the same coefficients. 

In this paper, the collected GCFs from the ball and heel were processed using the proposed method, which are collectively pictured in Figure 8a,b. The starting flags of on-ground status are marked as a triangle, while those of off-ground status are marked as a circle. It could be clearly illustrated in Figure 8a that more than one starting flag of on-ground status could be obtained in one gait cycle. However, this situation would not cause any detection error because the results were still identified as on-ground statuses. When the status divisions of ball and heel were accomplished, the gait pattern could be distinguished according to the rules, as shown in Figure 9a,b.

We apply the same CSMs to the gait phase detection of biped walking, and the results at different walking speeds are shown in Figure 10.

As can be seen from Figure 10, with the increase of walking speed, the GCF value gradually increases. The traditional fixed threshold method has great limitations. Our method can be well adapted to gait detection at different walking speeds. At the same time, we can adapt to the gait detection of the other foot by using only one foot data for training.

### 3.2. Accuracy and Reliability

As the off-ground and on-ground statuses were both identified using the two CSMs, the detection error rate should be calculated for both statuses. Note that *E*_1_ is the detection error rate for off-ground statuses, *E*_2_ is the detection error rate for on-ground statuses, *E*_3_ is the total detection error rate, and *E*_4_ is another detection error rate named the total data point error rate. As depicted in Table 4, the proposed method gains highly low detection error rates when compared with the reference method.

As can be seen from Table 4, the accuracy of the other subjects for testing is very high according to our evaluation protocol, except for subjects #16, #21, and #24. Even so, we find that with the same *E*_4_ protocol statistics, the total error rate of several high error rate subjects is less than 10%. This shows that our proposed model can be well adapted to the traditional way of statistics, with a strong fault-tolerant performance.

According to the evaluation protocol, *E*_1_, *E*_2_, and *E*_3_ for subject #5 were zero, indicating that the trained model could accomplish the gait classification perfectly. The corresponding *E*_4_ is 6.86% because of the difference in gait switching positions caused by the two different evaluation protocols. Furthermore, repeating the above-mentioned experiment ten times with 25, 50, and 75 samples, we count the overall average rate as an assessment of system accuracy and reliability, as depicted in Table 5.

Form Table 5d, the total average detection error rates were 4.10% and 3.92% for the training and testing data, respectively. On the other side, both the PM and STTTA methods were used to compute the detection error rates by *E*_4_. As a result, with the same evaluation protocol, 7.75%, 10.82%, and 10.55% were acquired for the CSM, the PM, and the STTTA, respectively. Experimental results demonstrated that the proposed method performed better in the application of gait pattern detection than the methods in the literature [25,26].

For each training model, the generated model parameters were different, and the maximum error rate using different protocol statistics was less than 10%, which showed that the proposed method was reliable and effective. Of course, choosing the optimal model to further reduce the gait detection error rate would be the next step in the future.

### 3.3. Robustness and Stability

As the on-ground and off-ground statuses were both identified using the CSM, the detection error rate should be computed for both status robustness.

For the second experiment, ten repetitive training sessions were conducted to calculate the error rate of each result, and the results were arranged from small to large according to *E*_3_ and *E*_4_ values, as shown in Figure 11.

As seen in Table 6, according to the error rate *E*_4_ statistics, the worst case is better than the other two methods, while the model training only uses GCF data of one subject at different speeds.

Obviously, the more data are used for training, the higher the accuracy of the model would be, and the better the model adapts to the changes of GCF at different speeds with strong robustness. The Table 6 indicates that the CSMs can be trained by only one subject and tested by other subjects of different weights.

### 3.4. Generalization Performance

Using the above-mentioned trained models, we tested and verified ten people walking for 60 s. Once the model had been trained, in real-time gait detection, the model’s input depended only on the current and past four sampling points of the sole GCF. The two CSM models respectively detect the out-of-synchronization flag events, and the detection results start with a new gait phase. Therefore, this model was fully applicable to real-time gait detection. Similarly, the global Lopez method is combined with filtering to label the data, and the comparison is performed by PM, STTA, and the CSM method proposed. The experimental results are as follows Table 7.

It can be seen in Table 7 that the detection result of #10 is the best. In evaluation mode 1, the accuracy of the gait detection result is 100%, and the detection result of #2 is the worst. Figure 12 and Figure 13 show the results of all gait tests.

It can be seen in Figure 12, Figure 13 and Figure 14 that in the real-time gait detection, the CSM method can obtain better real-time detection results without the new training model. The more training data, the higher the accuracy of the model detection, and the more versatile it is.

As a machine learning method, it was important to train data with quality and quantity. We chose multiple model trainings for different sizes of sample files. Although each trained model parameter was inconsistent, we achieved good accuracy on the testing set as shown in Figure 12, Figure 13 and Figure 14, which indicates that the proposed model had good generalization performance.

### 3.5. Advantages of the Research

When compared with the reference method, the proposed method could identify the gait patterns in real time, and obtain lower detection error rates than the methods [27,28] in the literature. If the CSM is trained well, it is not necessary to determine the threshold separately. The curve model contains a change process of plantar GCFs for a period of time, which has higher accuracy and robustness.

For a healthy human, gait pattern detection could be used to evaluate walking conditions. For people with foot diseases, traditionally, gait pattern detection could be used as quantitative data along with other temporal parameters (i.e., stride width, walking speed, cadence, and walking symmetry) to diagnose and prescribe patients’ pathological gaits and evaluate walking conditions after rehabilitation [23]. Meanwhile, gait pattern detection results would be used in several walking-aid devices, such as exoskeleton robots and smart wearable intelligent devices, or powered prostheses to help people with walking disabilities. 

Gait pattern detection plays an extremely important role to recognize the user’s intentions in the applications of exoskeleton robots and powered prostheses. Because the foot contacting the ground is directly related to classifying the gait patterns, force sensor platforms are commonly utilized with exoskeleton robots and powered prostheses.

### 3.6. Limitation of the Research

In our model, the template curve needs to be trained, and the training time of the model is long. Once the sensor changes, the model needs to be retrained, and it is difficult to adapt quickly. In our research, only healthy subjects were adopted on flat ground, not taking the pathological subjects into account. The experiments were done on a treadmill because the method is not suitable for irregular terrain and stairs walking.

## 4. Conclusions

This paper presents a curve similarity model as a classifier for real-time gait pattern detection. The curve is composed of a series of GCF data points, and the CSM is built offline to obtain a template curve. Then, the testing curve is compared with the template curve to figure out the similarity distance. The result of the computed similarity distance would lead to the division of off-ground and on-ground statuses by comparing the GCF with a given threshold. Finally, the gait patterns could be differentiated according to the detection rules. Under the same evaluation protocol, the proposed CSM acquired the lowest average detection error rates of 7.75%, when compared with PM (i.e., 10.82%) and STTTA (i.e., 10.55%). The experimental results demonstrated that the proposed method performed better in the application of gait pattern detection than the methods in the literature. Moreover, the proposed method could be used for real-time gait pattern detection and to detect the gait patterns adaptively.

## Figures and Tables

**Figure 1 sensors-19-03235-f001:**
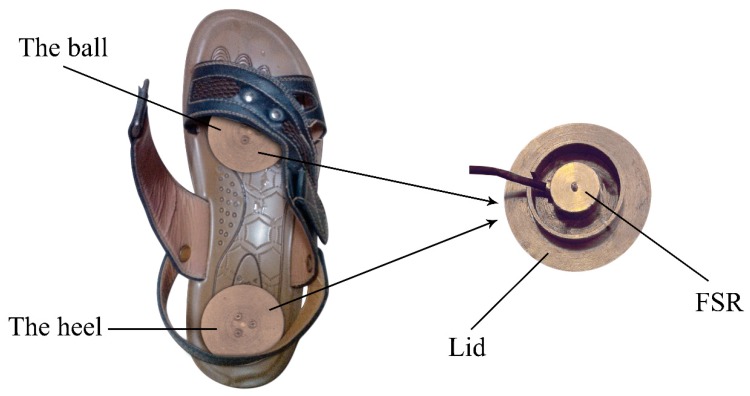
Two load cell mounted severally in the ball and heel, and a lid is used to enlarge the contact area.

**Figure 2 sensors-19-03235-f002:**
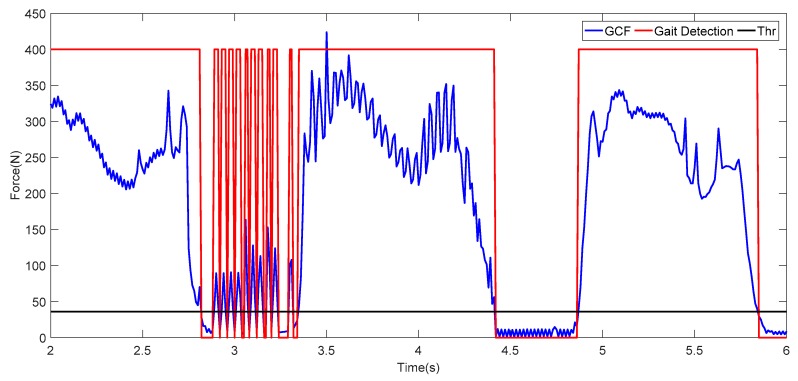
Abnormal values existing in the data collection for ground contact force (GCF). Thr, threshold.

**Figure 3 sensors-19-03235-f003:**
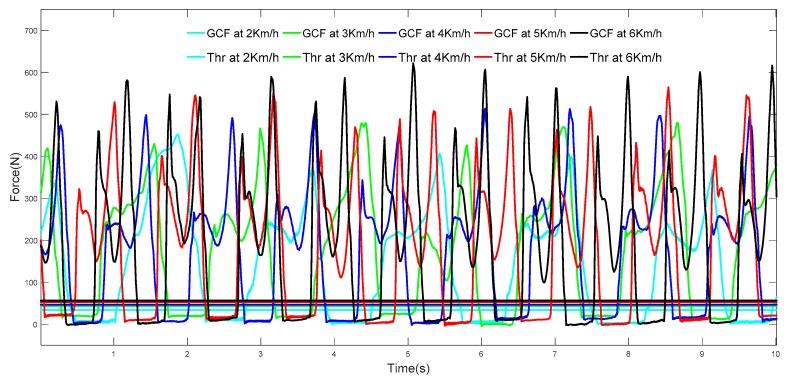
Change in GCFs with walking speed change.

**Figure 4 sensors-19-03235-f004:**
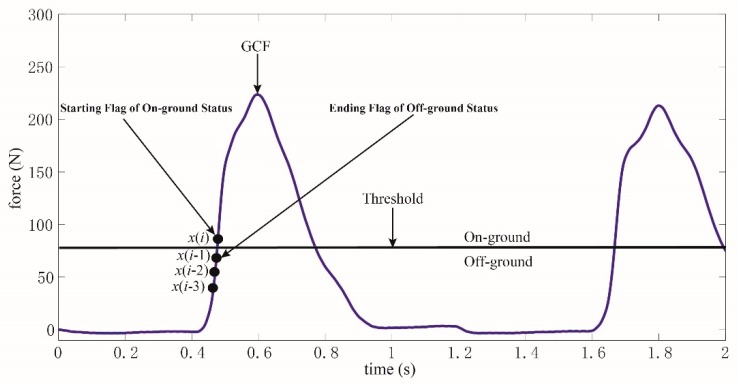
A curve consisting of four GCF points is used to identify the starting flag of on-ground status or the ending flag of off-ground status.

**Figure 5 sensors-19-03235-f005:**
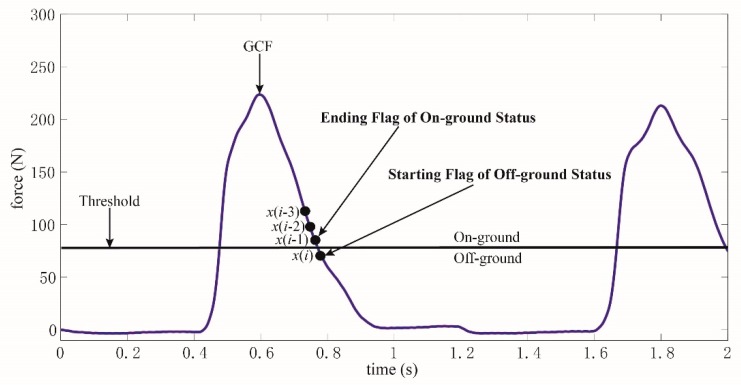
A curve consisting of four GCF points is used to identify the starting flag of off-ground status or the ending flag of on-ground status.

**Figure 6 sensors-19-03235-f006:**
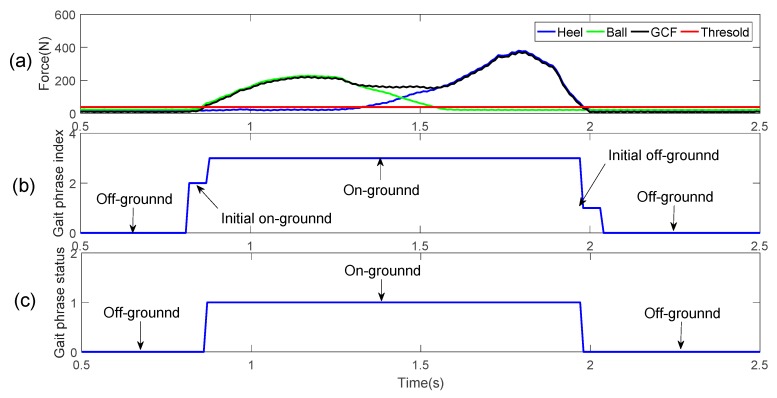
(**a**) Original gait GCF data, (**b**) four gait phase index, (**c**) two gait classification.

**Figure 7 sensors-19-03235-f007:**
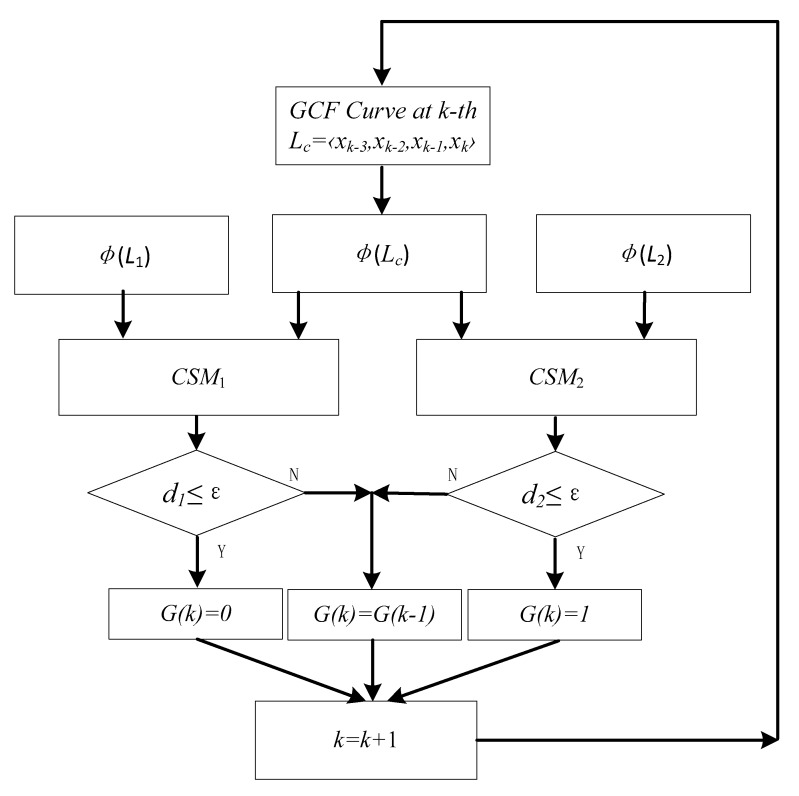
Flow chart of real-time gait detection algorithm. CSM, curve similarity model.

**Figure 8 sensors-19-03235-f008:**
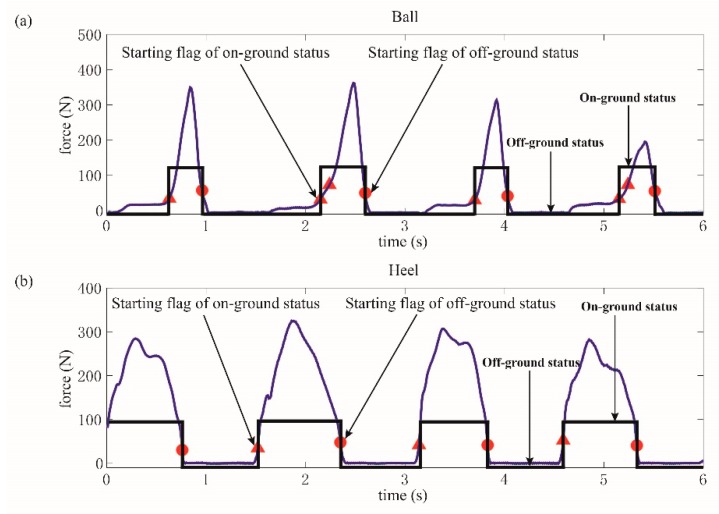
Status division for ball and heel. (**a**) GCFs and status division for the ball, (**b**) GCFs and status division for the heel.

**Figure 9 sensors-19-03235-f009:**
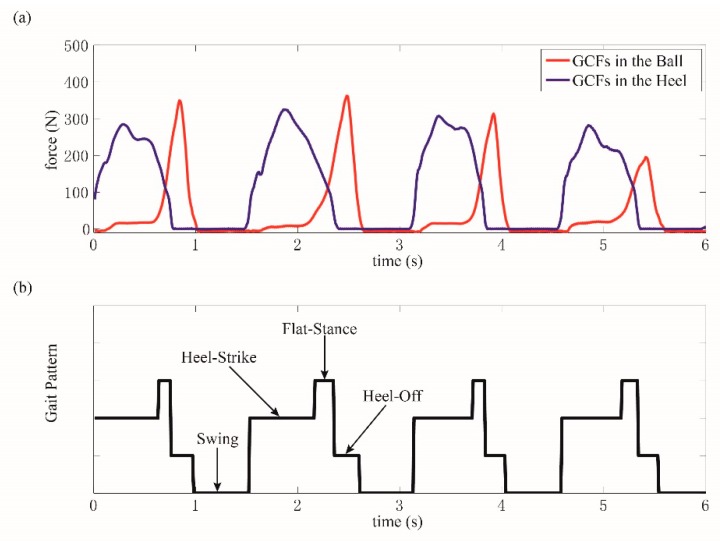
(**a**) GCFs in the ball and heel, (**b**) results of gait pattern detection.

**Figure 10 sensors-19-03235-f010:**
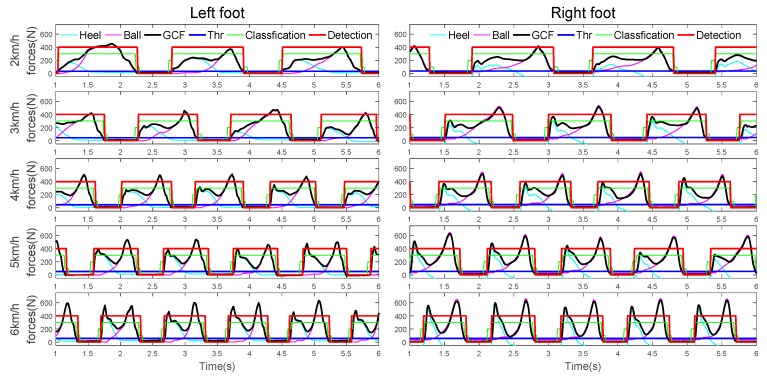
A subject gait phrase detection at speeds of 2~6 km/h.

**Figure 11 sensors-19-03235-f011:**
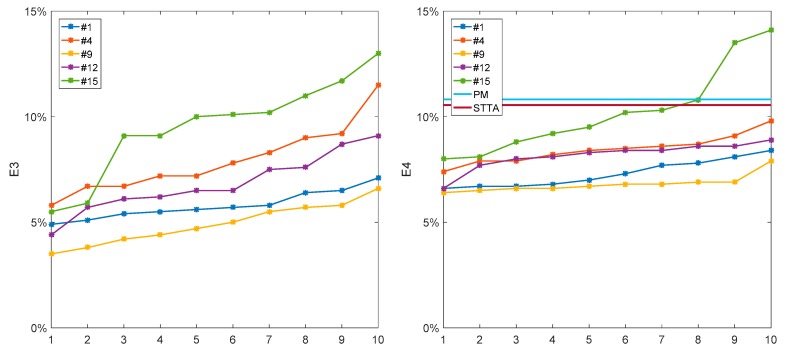
The average error rate of 10 times is counted, and the results are as follows. STTTA, self-tuning triple-threshold algorithm; PM, proportional method.

**Figure 12 sensors-19-03235-f012:**
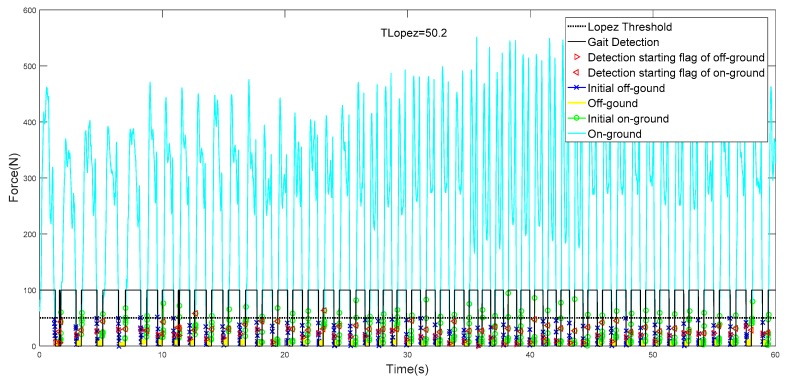
Result of gait phase detection.

**Figure 13 sensors-19-03235-f013:**
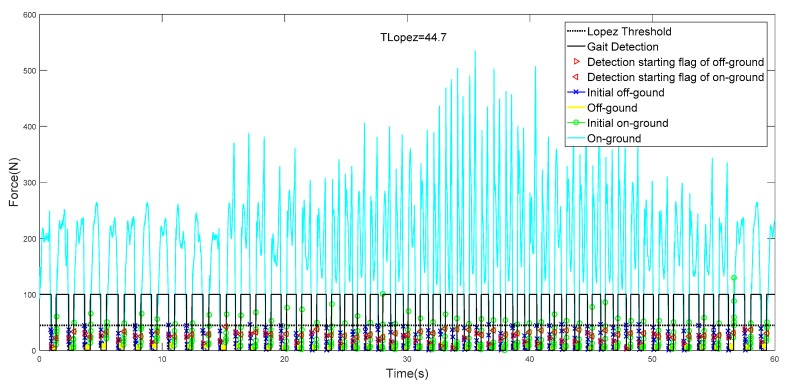
The best result of gait phase detection.

**Figure 14 sensors-19-03235-f014:**
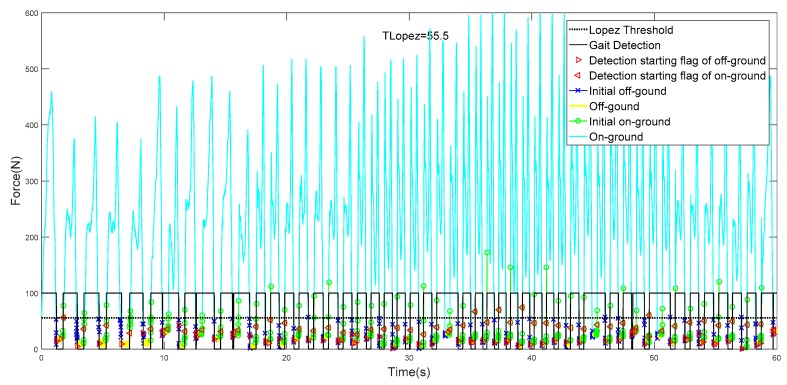
The worst result of gait phase detection.

**Table 1 sensors-19-03235-t001:** Evaluation of detection with true status.

Detection	True Status			
	On-Ground (3)	Initial Off-Ground (1)	Off-Ground (0)	Initial Off-Ground (2)
Starting flag of off-ground	×	√	√ *	×
Starting flag of on-ground	√ *	×	×	√
Other	Keep the last status			

**Table 2 sensors-19-03235-t002:** Parameters of curve similarity model (CSM) for off-ground status.

Order	1	2	3	4	5	6	7	8	9	10
*μ_j_*	53.11	−76.80	22.81	2.34	−26.25	−9.45	−2.47	−98.00	4.88	−68.36
*δ_j_*	19.82	112.80	51.29	45.95	57.44	54.45	29.67	77.87	24.00	37.81

**Table 3 sensors-19-03235-t003:** Parameters of curve similarity model for on-ground status.

Order	1	2	3	4	5	6	7	8	9	10
*μ_j_*	−7.27	32.13	8.23	74.03	−7.99	5.70	50.00	18.70	120.00	−3.50
*δ_j_*	162.78	68.21	52.75	33.85	39.12	31.11	61.04	32.84	33.71	107.32

**Table 4 sensors-19-03235-t004:** Detection error rates of CSM compared with the reference method by training and testing sets.

Subject	Training				Subject	Testing			
	*E* _1_	*E* _2_	*E* _3_	*E* _4_		*E* _1_	*E* _2_	*E* _3_	*E* _4_
#1	0.93%	4.72%	2.82%	6.04%	#11	0.77%	0.77%	0.77%	6.50%
#2	0.87%	5.22%	3.04%	6.07%	#12	0.86%	4.39%	2.61%	6.09%
#3	0.00%	0.00%	0.00%	6.86%	#13	0.93%	1.90%	1.41%	6.96%
#4	0.00%	8.33%	4.17%	7.67%	#14	0.93%	0.00%	0.47%	6.91%
#5	0.00%	0.00%	0.00%	7.05%	#15	3.33%	0.00%	1.67%	6.62%
#6	0.81%	0.00%	0.40%	6.73%	#16	38.89%	0.00%	19.44%	8.63%
#7	0.00%	3.70%	1.85%	6.45%	#17	2.17%	0.00%	1.08%	7.73%
#8	1.01%	0.00%	0.51%	6.17%	#18	0.00%	0.79%	0.40%	7.15%
#9	0.00%	5.15%	2.59%	8.57%	#19	3.17%	0.00%	1.59%	7.61%
#10	2.83%	5.71%	4.27%	7.64%	#20	3.60%	7.27%	5.43%	7.58%
Average	0.65%	3.28%	1.97%	6.93%	#21	16.42%	3.03%	9.77%	9.15%
					#22	0.00%	0.00%	0.00%	6.57%
					#23	0.00%	7.69%	3.85%	6.49%
					#24	13.04%	4.35%	8.70%	8.87%
					Average	8.06%	2.31%	5.19%	7.64%

**Table 5 sensors-19-03235-t005:** Detection error rates of CSM compared with the reference method. (**a**) Twenty-five samples were trained and the rest of the samples were tested and repeated 10 times; (**b**) 50 samples were trained and the rest of the samples were tested and repeated 10 times; (**c**) 75 samples were trained and the rest of the samples were tested and repeated 10 times; (**d**) average statistics of 25, 50, and 75 training samples and testing results. STTTA, self-tuning triple-threshold algorithm; PM, proportional method.

(**a**)								
**No.**	**Training**					**Testing**		
	***E*_1_**	***E*_2_**	***E*_3_**	***E*_4_**	***E*_1_**	***E*_2_**	***E*_3_**	***E*_4_**
1	0.35%	3.30%	1.82%	7.46%	6.53%	5.10%	5.82%	8.96%
2	0.52%	2.95%	1.74%	6.70%	2.54%	4.10%	3.32%	7.65%
3	0.17%	3.47%	1.82%	7.16%	3.45%	3.32%	3.38%	7.83%
4	0.35%	3.99%	2.17%	7.34%	4.31%	2.60%	3.45%	7.85%
5	0.52%	3.99%	2.26%	6.82%	4.76%	2.60%	3.68%	7.34%
6	0.35%	4.69%	2.52%	7.23%	2.86%	4.64%	3.75%	7.94%
7	0.17%	4.86%	2.52%	7.19%	4.13%	3.51%	3.82%	7.91%
8	0.35%	3.99%	2.17%	6.83%	3.04%	3.32%	3.18%	7.39%
9	0.35%	3.13%	1.74%	7.39%	8.30%	4.37%	6.34%	8.13%
10	0.35%	5.03%	2.69%	6.98%	3.94%	2.69%	3.32%	7.46%
Average	0.35%	3.94%	2.15%	7.11%	4.42%	3.44%	3.93%	7.75%
(**b**)								
**No.**	**Training**					**Testing**		
	***E*_1_**	***E*_2_**	***E*_3_**	***E*_4_**	***E*_1_**	***E*_2_**	***E*_3_**	***E*_4_**
1	0.43%	6.62%	3.52%	8.89%	6.39%	6.85%	6.62%	9.00%
2	1.06%	7.69%	4.37%	7.82%	2.08%	3.36%	2.72%	6.78%
3	1.28%	8.12%	4.69%	7.55%	2.63%	5.53%	4.08%	6.86%
4	1.06%	6.62%	3.84%	6.37%	2.49%	3.36%	2.92%	6.39%
5	2.34%	5.34%	3.84%	8.23%	5.75%	5.31%	5.53%	7.77%
6	0.85%	4.49%	2.67%	7.21%	4.08%	3.76%	3.92%	7.70%
7	0.64%	8.76%	4.69%	8.30%	2.36%	5.94%	4.15%	7.65%
8	0.64%	8.76%	4.69%	8.30%	2.36%	5.94%	4.15%	7.65%
9	0.85%	9.40%	5.12%	8.18%	1.72%	5.58%	3.65%	6.76%
10	0.64%	9.62%	5.12%	8.72%	2.45%	5.99%	4.21%	8.05%
Average	0.98%	7.54%	4.26%	7.96%	3.23%	5.16%	4.20%	7.46%
(**c**)								
**No.**	**Training**					**Testing**		
	***E*_1_**	***E*_2_**	***E*_3_**	***E*_4_**	***E*_1_**	***E*_2_**	***E*_3_**	***E*_4_**
1	0.55%	4.51%	2.43%	7.14%	9.38%	4.74%	7.06%	8.23%
2	0.69%	2.63%	1.66%	7.43%	1.54%	5.27%	3.41%	8.22%
3	0.64%	3.44%	2.03%	7.44%	3.74%	2.20%	2.97%	8.21%
4	0.96%	3.01%	1.98%	7.13%	9.93%	2.86%	6.49%	7.99%
5	0.69%	2.79%	1.74%	7.36%	1.76%	2.42%	2.09%	8.00%
6	0.59%	3.65%	2.11%	7.07%	2.09%	4.84%	3.47%	7.86%
7	0.64%	3.65%	2.14%	7.75%	6.83%	3.08%	4.95%	8.11%
8	0.85%	2.74%	1.79%	6.82%	4.63%	2.42%	3.52%	7.67%
9	0.69%	3.71%	2.19%	7.62%	1.87%	1.87%	1.87%	8.10%
10	1.12%	3.17%	2.14%	7.45%	4.63%	1.76%	3.19%	8.06%
Average	0.74%	3.33%	2.02%	7.32%	4.66%	3.15%	3.90%	8.05%
(**d**)								
**Sample Size**	**Training**					**Testing**		
	***E*_1_**	***E*_2_**	***E*_3_**	***E*_4_**	***E*_1_**	***E*_2_**	***E*_3_**	***E*_4_**
25	0.35%	3.94%	2.15%	7.11%	4.42%	3.44%	3.93%	7.75%
50	0.98%	7.54%	4.26%	7.96%	3.23%	5.16%	4.20%	7.46%
75	0.74%	3.33%	2.02%	7.32%	4.66%	3.15%	3.90%	8.05%
Average	0.69%	4.94%	2.81%	7.46%	4.10%	3.92%	4.01%	7.75%
Comparison with PM [24]								10.82%
Comparison with STTTA [25]								10.55%

**Table 6 sensors-19-03235-t006:** Detection error rates of CSM compared with the reference method by training and testing ten times with five samples.

No.	*E* _1_	*E* _2_	*E* _3_	*E* _4_
#1	2.71%	8.95%	5.79%	7.31%
#4	7.32%	8.82%	8.07%	8.44%
#9	2.75%	7.40%	5.07%	6.81%
#12	5.28%	8.47%	6.87%	8.34%
#15	7.90%	12.09%	9.99%	10.50%
Comparison with PM [24]				10.82%
Comparison with STTTA [25]				10.55%

**Table 7 sensors-19-03235-t007:** Detection error rates of CSM compared with the reference method with 10 samples under variable speed walking.

Sample	*E* _1_	*E* _2_	*E* _3_	*E* _4_
#1	2.13%	19.15%	10.64%	6.25%
#2	12.00%	26.53%	19.19%	10.23%
#3	7.55%	16.98%	12.26%	8.02%
#4	8.00%	2.00%	5.00%	5.88%
#5	0.00%	3.70%	1.83%	6.90%
#6	2.13%	0.00%	1.08%	5.53%
#7	1.89%	9.62%	5.71%	6.63%
#8	4.17%	31.25%	17.71%	9.60%
#9	1.96%	7.84%	4.90%	6.98%
#10	0.00%	0.00%	0.00%	7.17%
Average	3.98%	11.71%	7.83%	7.32%
Comparison with PM [24]				10.82%
Comparison with STTTA [25]				10.55%

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
