# Peer review of "Curve Similarity Model for Real-Time Gait Phase Detection Based on Ground Contact Forces"

_sensors, 2019, doi:10.3390/s19143235_

Round 1

Reviewer 1 Report

The proposed method is interesting. The authors have made many efforts to improve the quality of the paper, especially for the experiment part. However, with regarding to the newly added part of experiments, serious problems have been observed. Thus, the work is not fit for publication at its current form.

1) In the experiments, for training, only 50 samples are used. This is very small a dataset for training. It is not enough to validate the effectiveness of the proposed method. The authors are suggested to enlarge the dataset to evaluate the proposed method. 

2) The conclusion of the paper should be constrained. The conclusion of the whole paper should be supported by the experimental results.

3) Gait recognition is a hot research topic recently, and many innovative works have been published in the past several years. The most related and newly published works should be included in the literature review, e.g.,

Bei, S., Zhen, Z., Xing, Z., Taocheng, L., & Qin, L. (2018). Movement disorder detection via adaptively fused gait analysis based on Kinect sensors. IEEE Sensors Journal18(17), 7305-7314.

Rida, I., Almaadeed, N., & Almaadeed, S. (2018). Robust gait recognition: a comprehensive survey. IET Biometrics8(1), 14-28.

Zou, Q., Ni, L., Wang, Q., Li, Q., & Wang, S. (2018). Robust Gait Recognition by Integrating Inertial and RGBD Sensors.IEEE transactions on cybernetics48(4), 1136.

Tian, Y., Wei, L., Lu, S., & Huang, T. (2019). Free-view gait recognition. PloS one14(4), e0214389.

4) A response letter with point-to-point response to the commments should be uploaded along with the revised submission.

Author Response

Dear Sir,

Thank you for your comments for the manuscript. According with your advices, we amended the relevant part in manuscript. Here are my responses to the reviewers’ comments.

The proposed method is interesting. The authors have made many efforts to improve the quality of the paper, especially for the experiment part. However, with regarding to the newly added part of experiments, serious problems have been observed. Thus, the work is not fit for publication at its current form.

Comment 1: In the experiments, for training, only 50 samples are used. This is very small a dataset for training. It is not enough to validate the effectiveness of the proposed method. The authors are suggested to enlarge the dataset to evaluate the proposed method.

Response: Each sample file actually contains 3000 sampling point data, and the corresponding model training is 2997 (3000-4+1) lines of training or test data. Therefore, our experimental data (calculated by row) actually has a total of 360,000 samples, and here for the sake of visual description, we take the data file obtained by one person walking for 30s as a whole sample file. In the new experimental session, we have described this issue.

Comment 2: The conclusion of the paper should be constrained. The conclusion of the whole paper should be supported by the experimental results.

Response: We redesigned the experimental link, and designed three experimental links for the two databases to verify the accuracy, robustness and generalization performance of the model. As a machine learning method, it is important to train data quality and quantity. We chose multiple model trainings for different sample files. Although the model parameters of each training were inconsistent, we achieved good accuracy on the verification dataset, which indicates that our model has good applicability. 

Moreover, We had revised the conclusion as required.

Comment 3: Gait recognition is a hot research topic recently, and many innovative works have been published in the past several years. The most related and newly published works should be included in the literature review, e.g.,

Response: We have done that as required.

I have revised the manuscript accordingly and the revised portion is marked in red. I hope this will make it more acceptable for publication.

Yours sincerely,

Lie Yu

Reviewer 2 Report

Authors have made all the necessary changes in the article. The current version looks much better and more fluid, than the previous versions.

Author Response

Dear Sir,

Thank you for agreeing with that our resubmitted manuscript looked much better and more fluid than the previous versions.

Moreover, we had check the manuscript carefully.

Yours sincerely,

Lie Yu

Round 2

Reviewer 1 Report

The authors have made adequate improvements to the manuscript. My previous concerns have been addressed. I recommend an acceptance at this stage.

This manuscript is a resubmission of an earlier submission. The following is a list of the peer review reports and author responses from that submission.

Round 1

Reviewer 1 Report

This paper studied gait recognition using ground contact forces (GCF). Different from the traditional gait data such as video sequences, inertial time series or skeleton dynamics, the GCF reflects the gait dynamics in a perspective of the force changing on the foot. This data would be helpful for gait representation. This research is interesting, the proposed method is sound, which would give much inspiration to the related research in the community. However, several major concerns have been with the current version:

(1) The experiments were not well described in the manuscript. For example, how many data are collected on each subject? How many data are used for training or testing? Also, the experimental setting should be introduced with details. 

(2) The evaluation of the method should be well elaborated. For example, how many metrics have been used for the evaluation. The authors are suggested to refer to the evaluation protocols in Robust gait recognition by integrating inertial and RGBD sensors, IEEE Trans. on cybernetics, 2018. 

(3) The English writing should be improved. Many errors can be spotted in the manuscript, e.g., line 13, "realize the detection of specific gait patterns based the detection rules", line 66 "change", Line 68 " and the GCF change in a time 68 interval is taken as a curve to study."; all the equations shoul d be given punctuations, etc. 

Author Response

Dear Sir:

Thank you for your comments for the manuscript. According with your advices, we amended the relevant part in manuscript. Here are my responses to the reviewers’ comments.

This paper studied gait recognition using ground contact forces (GCF). Different from the traditional gait data such as video sequences, inertial time series or skeleton dynamics, the GCF reflects the gait dynamics in a perspective of the force changing on the foot. This data would be helpful for gait representation. This research is interesting, the proposed method is sound, which would give much inspiration to the related research in the community. However, several major concerns have been with the current version:

Comment 1: The experiments were not well described in the manuscript. For example, how many data are collected on each subject? How many data are used for training or testing? Also, the experimental setting should be introduced with details. 

Response: We have described the revision of the paper.

Comment 2: The evaluation of the method should be well elaborated. For example, how many metrics have been used for the evaluation. The authors are suggested to refer to the evaluation protocols in Robust gait recognition by integrating inertial and RGBD sensors, IEEE Trans. on cybernetics, 2018. 

Response: We describe in detail the use of E1, E2, E3 and E4 for two different evaluation indicators, as seen in formula (10) and (11).

Comment 3: The English writing should be improved. Many errors can be spotted in the manuscript, e.g., line 13, "realize the detection of specific gait patterns based the detection rules", line 66 "change", Line 68 " and the GCF change in a time 68 interval is taken as a curve to study."; all the equations should be given punctuations, etc. 

Response: Thank you for reading our paper so carefully, and we apologize for carelessness. We revise the whole paper as required.

I have revised the manuscript accordingly and the revised portion is marked in red bold. I hope this will make it more acceptable for publication.

Yours

Lie Yu

Reviewer 2 Report

The device shown/used in this study is a load cell, not FSR. Authors should use the right term. 

The significance of the study is not clear. it is much easier to just use the thresholding method, which has been the standard in gait analysis and to accommodate the different body weight, normally a certain percentage of the body weight will be used. It is easier to implement and widely accepted in the scientific community. Why would someone implement a more complicated algorithm just to detect the gait events? 

How many samples were analysed in this study?

What does it mean by comparing the results with gaitrite? Are the authors using gaitrite here? It is not clear how did the authors benchmark their study. Are they using instrumented treadmill? If yes, please specify clearly the model used, and some of the basic specifications.  

What are the timing errors? What are the errors between events detected by the algorithm and the benchmark? 

Authors claimed that their approach can be used in real-time, but there is no indication/results/description on how this can be used in real-time, especially, when the model requires several data points to detect on-off status. 

Author Response

Dear Sir:

Thank you for your comments for the manuscript. According with your advices, we amended the relevant part in manuscript. Here are my responses to the reviewers’ comments.

Comment 1: The device shown/used in this study is a load cell, not FSR. Authors should use the right term. 

Response: We revise the whole paper as required.

Comment 2: The significance of the study is not clear. it is much easier to just use the thresholding method, which has been the standard in gait analysis and to accommodate the different body weight, normally a certain percentage of the body weight will be used. It is easier to implement and widely accepted in the scientific community. Why would someone implement a more complicated algorithm just to detect the gait events? 

Response: Threshold method can not effectively detect walking for one subject of different speeds and weights, and the real-time performance is not strong, where it can be seen from the results in Fig.7. In the revised paper, we only use the walking data training model of one subject at different speeds for verification by all the subjects. The experimental results illustrate the effectiveness of our model. It is more robust than traditional threshold methods. In addition, our trained model directly corresponds to a classifier output, no need to determine the threshold, where the threshold has been given in advance, as seen in formula(8) .

Comment 3: How many samples were analyzed in this study?

Response: 120 samples of twenty-four subjects at five different speed analyzed in this study.

Comment 4: What does it mean by comparing the results with gait rite? Are the authors using gait rite here? It is not clear how did the authors benchmark their study. Are they using instrumented treadmill? If yes, please specify clearly the model used, and some of the basic specifications. 

Response: In this paper, we want to choose a method which had been proved to be effective in threshold computation. We had described that “As reported in [6], the Lopez-Meyer method had been approved through comparing itself with the “GAITRite system”, and gained a comparative and reliable confidence of 95%.” That is why we choose Lopez-Meyer method.

Lopez-Meyer method is a reference method used to compute a threshold for GCF division. This proposed method didnt use the threshold-based method for gait pattern detection, and the Lopez-Meyer method is used to test our method to determine the reliability.

There is no need to use the “GAITRite system” to verify our study, and we just need a reliable method to compute a threshold.

Comment 5: What are the timing errors? What are the errors between events detected by the algorithm and the benchmark? 

Response: If the GCF points at one sampling period had been collected, we processed the GCF data using two methods, such as the proposed method and the reference method. Both the two methods could differentiate the gait pattern. It is hypothesized that the gait pattern detected by the reference method is correct. If the gait pattern detected by the proposed method is equal to that detected by the reference method, there is no error. Otherwise, the error happened.

There is no timing errors needed to be computed in this paper. The gait events had happened and the data had been collected, and all we should do is to detect or recognize the gait pattern, and compare the detection result with the reference method.

Comment 6: Authors claimed that their approach can be used in real-time, but there is no indication/results/description on how this can be used in real-time, especially, when the model requires several data points to detect on-off status. 

Response: Our model is a classifier. The trained classifier can be used as a black box model directly. The difference between the current state and on-off state can be calculated directly by using formula (7). The current status can be judged directly by using the similarity threshold defined above. For example, if the threshold between on status and on status is less than the given threshold, then the current state is on status.

I have revised the manuscript accordingly and the revised portion is marked in red bold. I hope this will make it more acceptable for publication.

Yours

Lie Yu

Reviewer 3 Report

This work proposes a real-time gait phase detection using the curve similarity model and ground contact forces. The work is interesting for its research focus and applications, however the paper needs some explicit changes to improve the technical strength of the presented work.

1, Introduction section needs to focus more on the recent works published related to sensors for gait phase detection.

2, The choice made in using "ball" for the foot strike is not clear.,

3, Was the study conducted with a formal ethical approval?

4, Experiment procedure: Using treadmill and alternating the walking speed is acceptable, however the restricted experiment time for 30s and 5 trials is a very limited data to validate the method. The authors can at least introduce 2 or 3 sets of 5 trials with alternated speed or use the similar style as in rehabilitation: 10 metre walking test or 6 minutes walking test.

5, Evaluation of the results needs more explanation. Any specific reason behind choosing Lopez-Meyer method as reference? 

6, Results and Discussion section can be combined,based on the flow of the paper.

7, Discussion needs to be more explicit about the major scientific contribution and research questions of this work

Author Response

Dear Sir:

Thank you for your comments for the manuscript. According with your advices, we amended the relevant part in manuscript. Here are my responses to the reviewers’ comments.

This work proposes a real-time gait phase detection using the curve similarity model and ground contact forces. The work is interesting for its research focus and applications, however the paper needs some explicit changes to improve the technical strength of the presented work.

Comment 1: Introduction section needs to focus more on the recent works published related to sensors for gait phase detection.

Response: We had done that as required.

Comment 2: The choice made in using "ball" for the foot strike is not clear.,

Response: In this paper, the heel is for the foot strike detection.

Comment 3: Was the study conducted with a formal ethical approval?

Response: This work was supported by the National Key R&D Program of China The study on Load-bearing and Moving Support Exoskeleton Robot Key Technology and Typical Application, while Gait pattern detection is an important technology in this program. As the experiments had no harm to the subjects and only healthy people are invited to participate in this experiment, no formal ethical approval is made before the experiment.

Comment 4: Experiment procedure: Using treadmill and alternating the walking speed is acceptable, however the restricted experiment time for 30s and 5 trials is a very limited data to validate the method. The authors can at least introduce 2 or 3 sets of 5 trials with alternated speed or use the similar style as in rehabilitation: 10 metre walking test or 6 minutes walking test. 

Response: We will continue to do further research in the future.

Comment 5: Evaluation of the results needs more explanation. Any specific reason behind choosing Lopez-Meyer method as reference? 

Response: We had described that “As reported in [6], the Lopez-Meyer method had been approved through comparing itself with the “GAITRite system”, and gained a comparative and reliable confidence of 95%.” That is why we choose Lopez-Meyer method .

Comment 6: Results and Discussion section can be combined, based on the flow of the paper. 

Response: We had done that as required.

Comment 7: Discussion needs to be more explicit about the major scientific contribution and research questions of this work

Response: We had revise that as required.

I have revised the manuscript accordingly and the revised portion is marked in red bold. I hope this will make it more acceptable for publication.

Yours

Lie Yu

Round 2

Reviewer 2 Report

In response to previous comment 2, authors mentioned that: "Threshold method can not effectively detect walking for one subject of different speeds and weights, and the real-time performance is not strong, where it can be seen from the results in Fig.7." It is not very clear how the performance of the threshold method is not strong. Authors should describe this in details in the manuscript. In Fig 7, it can be seen that the "Thr" (I presume Thr is the threshold method) is flat. How can this be flat? 

It seems that Lopez method is quite crucial here, so please briefly describe Lopez method in the text too, so that the future readers have a glimpse on what this method is all about without having to search online for the details, especially considering that not all readers may have access to this article. 

There are several typographical errors and grammatical errors. Authors are suggested to revise the manuscript accordingly. 

Reviewer 3 Report

Authors have tried to answer most of concerns raised but some of the main questions are still not answered.
A lack of ethical approval can be considered as a serious flaw for most of the studies. Although, there is no
significant damage to be caused to the subjects based on the study design, a local approval from their university
is still required to ensure that the study was performed with the ethical regulations.

The experimental study design seems to be weak and definitely needs a serious consideration about the data. If the authors consider that this
small amount of data is significant to validate their study, then a statistical validation of the data is necessary.

Data comparison with an existing method in literature is acceptable., but how is this compared?

From literature, the gait cycle identification is based on Foot strike and Heel strike., there is no mention of ball strike. Foot strike
can be further classified as fore-foot strike or mid-foot strike. The use of ball strike instead of foot strike is still not clear.

Section 2.3: Authors have explained the process very clearly. What is TLopez? how do you calculate?

For the evaluation, how the refernce method validation is conducted? There is no mention about the validation in results section.

Results and discussion
1,Second paragraph is not clear. how the selection of specific samples happened? Why these samples and not others? any scientific reason?,
2, section 3.1, threshold calculated through Lopez-Meyer method, what is the threshold value?
3, From fig 7, the threshold is observed at a very low level., approx 50N is it right? How come the maximum force observed is in the similar
range in all the 5 different speed?

The paper needs to be reviewed by a native english speaker for grammatical and spelling errors.